# Cell-like pressure sensors reveal increase of mechanical stress towards the core of multicellular spheroids under compression

M.E. Dolega[1], M. Delarue[2], F. Ingremeau[1], J. Prost[2], A. Delon[1] & G. Cappello[1]

The surrounding microenvironment limits tumour expansion, imposing a compressive stress on the tumour, but little is known how pressure propagates inside the tumour. Here we present non-destructive cell-like microsensors to locally quantify mechanical stress distribution in three-dimensional tissue. Our sensors are polyacrylamide microbeads of well-defined elasticity, size and surface coating to enable internalization within the cellular environment. By isotropically compressing multicellular spheroids (MCS), which are spherical aggregates of cells mimicking a tumour, we show that the pressure is transmitted in a non-trivial manner inside the MCS, with a pressure rise towards the core. This observed pressure profile is explained by the anisotropic arrangement of cells and our results suggest that such anisotropy alone is sufficient to explain the pressure rise inside MCS composed of a single cell type. Furthermore, such pressure distribution suggests a direct link between increased mechanical stress and previously observed lack of proliferation within the spheroids core.

[1] Université Grenoble Alpes, Laboratoire Interdisciplinaire de Physique, CNRS, F-38000 Grenoble, France. [2] Institut Curie, CNRS, Université P. et M. Curie, UMR168, F-75231 Paris, France. Correspondence and requests for materials should be addressed to M.E.D. (email: monika.dolega@univ-grenoble-alpes.fr) or to G.C. (email: giovanni.cappello@univ-grenoble-alpes.fr).

An intriguing question, that remains unsolved, is how multicellular organisms that are so diverse in their final form are derived from the basic organizational group of cells at the origins. What cues determine cells fate within the forming tissue or during the initiation of a disease like cancer? Over decades of research on morphogenesis, multiple biochemical pathways responsible for embryo development progression were identified[1,2]. Interestingly, those pathways were also activated during tumour development, suggesting that tumorigenesis progresses through a reversed developmental program[3,4]. Multiple recent studies refocused on the role of mechanical cues in tissue morphogenesis and homeostasis, after pioneering works showing that not only biochemical signalling, but also mechanical stress is indispensable for example during Drosophila gastrulation[5,6] or neural tube extension in vertebrates[7]. Extensive in vitro studies on the mechanical cues (that is, ECM rigidity, application of a flow to induce shear stress), showed that these alone can promote malignant phenotype in a non-malignant cells[8] or promote proper three-dimensional (3D) growth and development of malignant cells[9]. There are many more examples of processes (not only during development), where presence of mechanical stress has been inferred from experimental approaches, such as birefringence measurements[10] or by observation of the geometry of the cell shapes in the tissue[11]. Despite these qualitative observations, which directly link cell behaviour with mechanical stimuli, the precise mechanisms by which mechanical forces affect crucial biological processes during development and tumorigenesis remains unknown.

One reason the progress in understanding the role of mechanical stress in tissue morphogenesis and homeostasis does not follow the pace of biochemical studies, comes from the lack of appropriate tools to measure forces in vivo or in vitro in 3D culture models. Cells constantly receive various tissue-associated physical forces including hydrostatic pressure, shear stress, compression and tension, and out of those we are able to measure only few. Measure of a cellular tension relies on the use of femtosecond-pulsed laser to ablate cell–cell junctions[12,13]. The tension is determined from the speed of the retraction of the cut junction and the measure is qualitative because the mechanical properties of the cells and their surrounding remain unknown. Micropuncture[14] and wick-in-needle technique[15] are used to define the interstitial hydrostatic pressure within tissues. To quantify anisotropic stresses within living tissues, Campas et al.[16] developed force transducers in the form of small oil microdroplets functionalized with ligands for cell surface receptors, which are injected in the intercellular space of tissue. Force measurement relies on the full confocal reconstruction of the droplets deformation from its spherical shape normally present at equilibrium. Owing to the lack of compressibility of oil droplets, this method is not applicable to define the isotropic component of the stress, which is expected to increase in developing tumours[17].

Growth of a tumour occurs through reactivated proliferation program of cells[3,4], which as a consequence induces compressive stress accumulation at the tumour–stroma interface. The first experimental approach to prove accumulation of mechanical stress during growth in constrained environment relied on embedding multicellular spheroids (classical in vitro model of tumour) within agarose gels of controlled mechanical properties[18]. Growth under such constrained conditions decreased cell proliferation and induced apoptosis. Moreover, further studies revealed that inhibition of cell proliferation upon externally applied stress is not uniform in multicellular spheroids and is reversible once the stress is released[19]. With the discovery of the mechano-sensitive YAP/TAZ pathway controlling cell proliferation and survival[20], it becomes crucial to define the distribution of isotropic stresses inside tumours grown under mechanical stress to understand emerging cellular phenotypic heterogeneity.

To circumvent present technical limitations, we introduce a novel approach that enables for the first time to directly and locally quantify mechanical stress in 3D. Our method includes fabrication of uniform and mechanically well-defined elastic polyacrylamide (PAA) microbeads, which when incorporated within the intercellular volume serve as internal cell-like sensors of mechanical stress. Fabricated polyacrylamide microbeads are (i) functionalized to promote cellular adhesion, (ii) loaded with a fluorophore to facilitate imagining, (iii) compressible and (iv) have homogenous elastic properties. Therefore, quantification of the local mechanical pressure within the tissue relies on defining the strain (change in volume) of integrated beads. Using our newly developed methodology, we observed and quantified the propagation of an externally applied isotropic stress within the multicellular spheroids of malignant murin colon cancer cells. Our measurements reveal that the mechanical stress is non-uniformly distributed, and that the stress profile is related to the cellular shape anisotropy.

## Results

**PAA microbeads as stress sensors.** Besides the compressibility, polyacrylamide gels present several important advantages making them one of the most often applied materials to study cell behaviour in two-dimensional cultures on soft substrates[21,22]. The elastic modulus of PAA gels can be tuned by changing the relative concentration of acrylamide to bisacrylamide[23]. Furthermore, PAA is biocompatible and inert, thus cellular adhesion relies only on the type of ligand chemically coupled to the surface. The pore exclusion size of PAA gels is small enough (tens of nm, ref. 24, Supplementary Note 1) as compared with other natural and synthetic hydrogels, so that cells and their protrusions do not enter the gel. Those properties made PAA gels ideal candidates for the fabrication of tissue pressure sensors.

To fabricate PAA microbeads we used an oil-in-water emulsion approach. By vortexing the two phases, a dispersion of water droplets was formed in the oil phase. The emulsion of acrylamide phase in oil was formed using perfluorinated oil HFE-7500 supplemented with PFPE-PEG surfactant[25]. Although the effect of the presence of oxygen was not as drastic in bulk preparations, we noticed that in case of small droplets where surface to volume ratio was high, polymerization in oxygen-reduced conditions was crucial. Once the emulsion was formed it was purged for 5 min with argon, and let to polymerize for 1 h in 60 °C to produce spherical microbeads (Fig. 1a). During the polymerization, we incorporated a fluorescent 500 kDa dextran-FITC (Fig. 1b) of size sufficiently large to remain physically trapped in the polyacrylamide gel after polymerization. The fluorophore was incorporated to enable fluorescent visualization of beads. PAA microbeads were also functionalized with fibronectin (Fig. 1c) to enable internalization of beads between cells. Finally, to verify how concentration of surfactant influences the mean size of microbeads, we used oils containing 1, 3 and 5% (w/w) of surfactant. We found that the fabrication method was stable and only little sensitive to the concentration of surfactant. Increase in surfactant concentration to 3 and 5% has provided about 5% higher fraction of small beads (~30% of beads were below 40 μm) as compared with 1% (Fig. 1d).

Homogenous and well-defined mechanical properties of the PAA cell-like sensors are required to quantify the isotropic

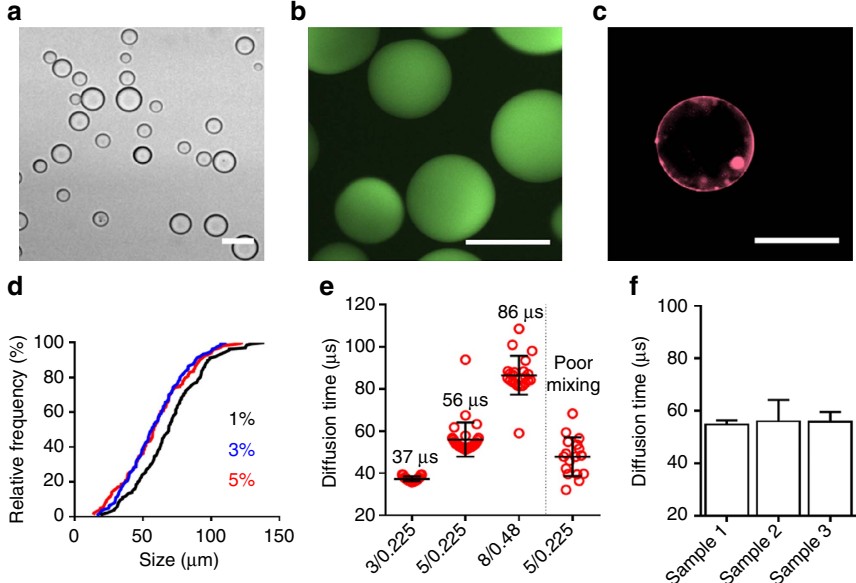

**Figure 1 | Characterization of polyacrylamide (PAA) beads.** (**a**) Bright-field image of polymerized PAA beads after filtration. Scale bar, 50 μm. (**b**) Fluorescence image of PAA microbeads containing trapped large polymers functionalized with FITC; scale bar, 50 μm. (**c**) Fluorescence image of coating of PAA beads with Cy3-Fibronectin; scale bar, 50 μm. (**d**) Distribution of size of polyacrylamide beads in dependance on the concentration of PFPE-PEG surfactant (1, 3 and 5%) during initial vortexing. (**e**) Characteristic diffusion time of SRB molecules, using FCS, within gels indicated uniformity of beads (small dispearsion) and mechanical properties. Small time of diffusion is characteristic for soft gels and increases with the volume fraction. Gels are defined by the acrylamide/bisacrylamide ratio. On the right, for 5/0.225 gels we show the effect of mixing on the uniformity of the batch. Each point corresponds to a single mesure of the diffusion time within the bead. (**f**) Uniformity and reproducibility is maintained between batches (sample 1, sample 2 and sample 3) of the same acrylamide/bisacrylamide ratio (5/0.225). (**e**,**f**) Mean ± s.d.

stress within tissues. To verify how uniform are beads within the batch and between batches prepared independently, we used fluorescence correlation spectroscopy (FCS) technique. For FCS measurements, an additional small and very mobile fluorescent molecule, sulphorhodamine B (SRB, hydrodynamic radius 0.5 nm (ref. 26)), was infused into a gel at a very low concentration (nM). Total intensity within the confocal spot was registered and temporal fluctuations were autocorrelated to determine the characteristic time of diffusion of molecules through the confocal spot. This time of diffusion, besides temperature and viscosity of the medium, depends on the gel characteristic parameters such as volume fraction[27] and pore size. We have confirmed that our method of fabrication was robust and reproducible between independently prepared batches of the same composition (Fig. 1f) and that the diffusion time depended on the composition of the PAA microbead and initial mixing efficiency (Fig. 1e).

We determined the bulk modulus of microbeads by recording the change in volume under isotropic stress. To do so, we applied a recently published method on the use of big molecules of dextran to induce osmotic stress[28]. We estimated that the exclusion pore size of fabricated PAA microbeads was smaller than ~10 nm by observing the diffusion of different fluorescent tracers. To ensure controlled osmotic stress applied externally we used a 2 MDa dextran. Since the hydrodynamic radius of 2 MDa dextran (27 nm) was much higher than the exclusion size of pores within the gel, the system continuously attempts to equilibrate and the resultant osmotic stress mechanically compresses PAA microbeads (Fig. 2, Supplementary Note 1). The applied pressure depended on the concentration of dextran, which has been previously calibrated and broadly described[29]. We have determined the stress/strain response of PAA microbeads by applying a range of pressures and observing the resultant volume decrease (Fig. 2). In an elastic region of deformation (compression below 15%),

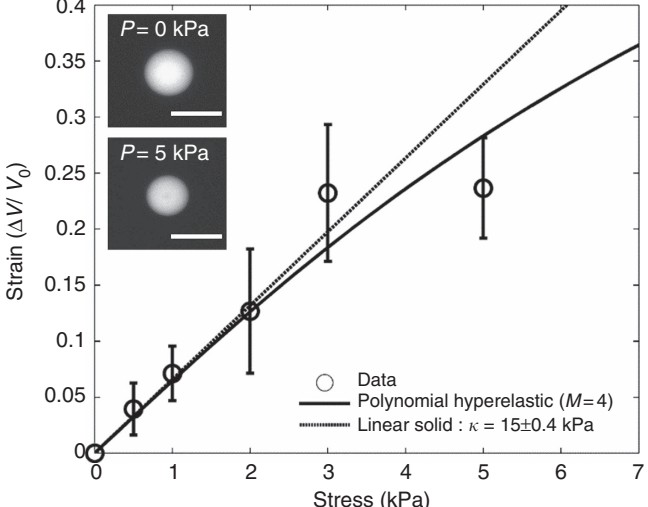

**Figure 2 | Determination of bulk modulus of PAA microbeads (5/0.225).** Stress–strain relation for PAA beads prepared by the ratio of 5/0.225 acrylamide/bisacrylamide. The linear fit has been determined for the first three points. We used and fitted polynomial Rivlin–Mooney model to determine the strain in the non-linear region. Results are represented by mean ± s.d. Inset: fluorescent images of a bead at the initial state $P = 0$ kPa and after application of a mechano-osmotic stress $P = 5$ kPa. Scale bar, 20 μm.

the stress is linearly proportional to the strain. To model mechanical properties of polyacrylamide beads out of linear regime, we used an empiric polynomial Mooney–Rivlin model. The continuous line in Fig. 2b corresponds to the best

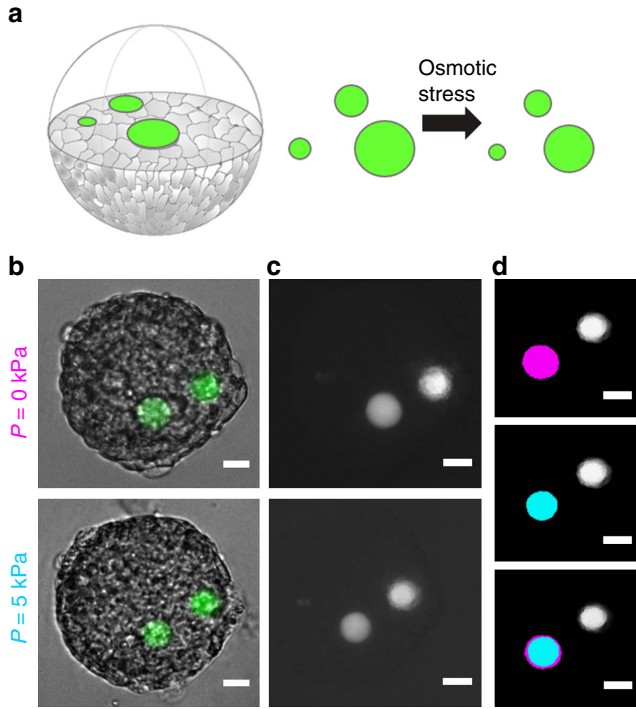

**Figure 3 | Experimental approach to measure pressure propagation.**
(**a**) Schematic ilustration presents the concept of the application of microbeads as pressure sensors within multicellular spheroids. Upon externally applied stress, embedded beads undergo compression as shown in **b,c**. Bright-field images (**b**) serve to the contour of spheroids at equatorial plane. (**c**) Fluorescence of beads have been used to determine the surface before and after applied stress. (**d**) Present masks created upon IsoData threshold. The difference of surface area (magenta represents state before and cyan represents state after compression) as shown by superimposed images is converted into the strain, $|V(P) - V_0|/V_0$, for each bead found in focus. Scale bar, 20 μm (**b-d**).

fit between the experimental stress/strain curve and the polynomial Mooney–Rivlin model. The bulk modulus of the beads (5/0.225 acrylamide/bisacrylamide ratio) used in all experiments was deduced from the slope at the origin of the stress/strain curve to be $15 \pm 0.4$ kPa (mean ± s.e.).

**Mechanical stress propagation in MCS upon compression.**
Multicellular spheroids (MCS) are spherical cell aggregates composed of hundreds to thousands of cells that are broadly used as *in vitro* tumour models in drug assays, proliferation and invasion studies[30]. In here, we used spheroids as model tissues to study the propagation of externally applied compressive stress within these cellular aggregates. To do so, we incorporated cell-like pressure sensors functionalized with fibronectin (radius $< 20$ μm) by introducing them among cells during the initial process of cellular aggregation (Supplementary Movie 1). We fabricated only small spheroids (radius $120$ μm $\pm 10$ (mean ± s.d.)) to avoid the occurrence of a necrotic core, which could drastically hamper our measurements. Formation of spheroids (CT26 cell line) typically took 2 days and successful incorporation of three to four beads per spheroid has been confirmed by epifluorescence microscopy in the majority of 3D structures.

To study the effect of compressive stress on spheroids, and more particularly assess the local stress, before-and-after measurements of beads size have to be performed on the same

spheroid and the same bead (Fig. 3a–c). Therefore, we have washed spheroids from the remaining non-incorporated beads and isolated them in multiwell plates. Continuous cellular and global spheroids movement hampered live 3D confocal imagining during the acquisition. To avoid such technical problems, as well as the necessity to reconstruct 3D images, we used epifluorescence microscopy to define the surface area of every spheroid and incorporated beads at the equatorial plane, which corresponded to the Z-position where particular objects were in focus (Fig. 3d). In rare cases we have encountered beads that were initially deformed into an elliptical shape (a fraction of $\sim 1/100$). We eliminated these beads from the analysis due to their occasional appearance and lack of clear explanation of their origin (that is, during the fabrication process). Volume of beads was calculated from the surface area considering that beads were perfectly spherical. Position of beads within spheroid was defined from the Z-coordinates of the bead and the spheroid (Z-focus, read-out from motorized microscopic stage), and X–Y coordinates of beads were defined from the images. Beads that were within 10 μm from the surface were ignored to avoid analysis on beads that could have been only partially embedded within the structure. Focus on spheroids and beads was adjusted manually. We verified that the error due to focus adjustment was negligible with objectives of low numerical aperture ($< 0.3$) and large depth of field ($> 6$ μm). (Supplementary Note 2). To define the surface area of beads we used ImageJ IsoData automatic method of threshold determination[31] performed on ROI around the bead (Supplementary Note 3).

In general, incorporated PAA beads were randomly distributed within the volume of spheroids, allowing us to define the pressure profile along the radius. A single spheroid contained at most three to four beads within the volume. Spheroids and beads were imaged before the application of compressive stress and 30 min after to ensure that system was at steady state. We applied a mechano-osmotic stress of $\Delta P_0 = 5$ kPa, which is in the physiological range of stresses occurring *in vivo* in growth-constrained environment[32]. Upon 5 kPa stress spheroids reduced their volume of $\sim 5\%$ after 30 min. The strain of beads has been measured by the relative difference of initial volume $V_0$ of the bead and the volume $V(P)$ after the application of the external stress: $|V(P) - V_0|/V_0$. If we hypothesize that spheroid is a homogenous viscoelastic sphere, stress and strain fields are expected to be uniform. Conversely, read-out from spatially distributed microbeads surprisingly indicated that the mechanical stress increases towards the centre and drops by the core of the spheroids (Fig. 4a, $N = 50$). Interestingly in the outer shell of the spheroid (70–80% of the distance from the core), the mean measured strain was $0.09 \pm 0.02$ (mean ± s.e.m.), which is approximately twice smaller than expected value for isolated beads at equivalent stress (mean strain $= 0.23$, $N = 7$). This strongly suggested a pressure jump at the spheroid surface from 5 kPa to $\sim 1$ kPa. Moreover, $\sim 60\%$ of all strain measurements within compressed spheroids have fallen within the linear region of deformation of microbeads (strain smaller than 20%). To obtain a quantitative measure of the stress along the radius of CT26 spheroids exposed to an isotropic compression, we have used the stress/strain calibration curve of microbeads (Fig. 4b). Our results suggest that the surface discontinuity acts as a screen to the external stress ($\Delta P_{\text{periphery}} < 1$ kPa $\ll \Delta P_0$). On the contrary, the strain progressively increased towards the spheroid core indicating that in the whole volume of the spheroid, measured stress was significantly lower than the externally applied pressure. Only by 30% of the distance from the core ($r/R_0$) measured mean strain was very close (and locally exceeded) to the one obtained for isolated beads under 5 kPa pressure.

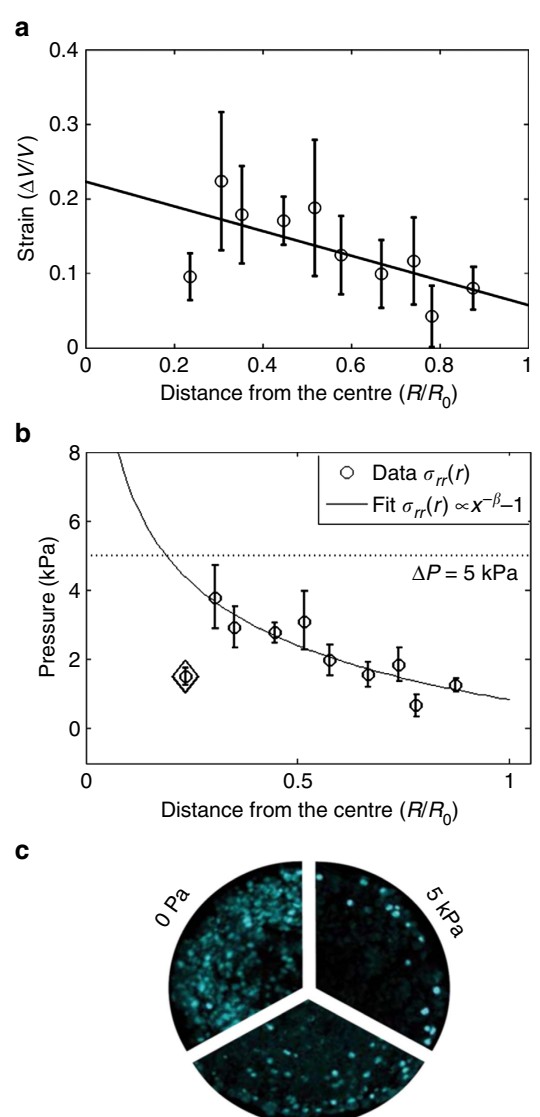

**Figure 4 | Pressure distribution in CT26 spheroids upon 5 kPa compressive stress.** (**a**) Strain profile of microbeads $|V(P)-V_0|/V_0$ along the radius of the spheroid. Data points are grouped together in bins, with the error bar being a s.e.m. and the position being an average position within the bin. The solid line is a linear fit to all experimental data points and it indicates an increasing strain towards the core of the spheroids. (**b**) Pressure profile obtained with a stress/strain calibration curve of PAA microbeads. The solid line represents a fit of $r^\beta$ with $\beta = -0.21 \pm 0.1$. Error bars are s.e.m. The point for $r \sim 0.22$ (losange) has been omitted in this fit. (**c**) Cellular proliferation along the radius in a control (0 Pa) and spheroids grown under 1 and 5 kPa pressure. Images represent immunostainning for Ki-67. Reprinted with permission from Montel *et al.*[33]

Our observation correlated well with the reduction of cell proliferation in the core of the spheroid put under isotropic compression. In Fig. 4c (reprinted from Montel *et al.*,[33]), we compare the distribution of the proliferating cells (in cyan, Ki-67 positive) in control spheroids and in spheroids under pressure. Applied compressive stress impeded cellular proliferation. With the increasing compressive stress the cell cycle of cells situated within the volume was hampered. Our previous observations revealed that the compressive stress inhibits cell proliferation by an overexpression of the kinase

inhibitor p27[Kip1] at the level of the Restriction point[19]. Surprisingly, proliferation continued in outer layers of the structure, where the measured local pressure increase was significantly lower ($P < 1$ kPa) than pressure applied externally ($\Delta P_0 = 5$ kPa). A continuous increase of pressure towards the centre of spheroids can explain lack of proliferation within the volume. This observation was compatible with the known anti-proliferative effect of mechanical stress[18,19] and indicated that cells actively responded to stress and that necrotic core formation is not necessarily caused only by the gradient of nutrients but can be in parallel enhanced by the unavoidable mechanical stress emerging from tumour growth.

**Stress distribution is correlated with cellular anisotropy.** Interestingly, observation of structural organization of CT26 spheroid revealed that the cell orientation strongly depends on the position within the spheroids. Figure 5a,b shows an optical section and quantification of the shape of the CT26 cells along the radius of the equatorial plane. We observed that only within the surface layer, cells represented a 'round' shape and were loosely organized, while within the structure cells appeared progressively compressed towards the core, in the radial direction, and stretched in the ortho-radial plane, with an aspect ratio larger than two.

Several studies confirm the role of the organization of the cytoskeleton components and polarity on the anisotropic response to stress[34,35]. Such structural anisotropy may lead to anisotropy in the mechanical response of the cells. We have inhibited ROCK, which is known to control cell cytoskeleton assembly and contractility, and we observed that pressure propagation was altered as compared with control (Supplementary Note 4). Interestingly, observed difference occurred in the region where cells in CT26 spheroids continue to divide. This suggests that the proliferation might be one of the factors imposing anisotropy within the cellular aggregates. Moreover, cells shape quantification revealed a discontinuity as the anisotropy decreased in the core of the aggregate (aspect ratio $\sim 1.5$), where cells become rounder (Fig. 5a,c). This is consistent with our previous observation that cell polarity radially depended on the position inside the spheroid (of BC52 cells), where the nucleus–centrosome axis was preferentially oriented towards the spheroids core (Supplementary Fig. 4). Furthermore, initially loose cellular organization at the surface underwent significant changes upon pressure jump of 5 kPa (Fig. 5d). Also beads trapped at the surface underwent high deformations, and represented a pear-like shape (Fig. 5e). Such pear-like shape is reminiscent of actin propelled liquid drops observed by Boukellal *et al.*[36] and suggests an active behaviour of the superficial cells in response to stress.

Following evidences of anisotropic cellular behaviour, Delarue *et al.*[37] proposed a model to describe the response of an anisotropic elastic sphere to an isostatic compression. Using this model, we compute the radial stress profile in three cases (see schema in Fig. 6: (a) the cells are softer in the radial direction than in the ortho-radial plane, (b) the cells are mechanically isotropic, (c) the cells are stiffer in the radial direction). The three profiles of the radial stress are plotted in Fig. 6 as a function of the distance from the spheroid centre, for a superficial stress of 1 kPa. Although the stress is constant in the isotropic case (Fig. 6b), it vanishes in the centre when the cells are softer in the radial direction than in the tangential one (Fig. 6a). This can be intuitively understood as an 'arching effect' with the outer layer bearing all the stress. Interestingly, the situation is reversed when the radial direction is the 'stiffest one' (Fig. 6c). In the latter case, the stress diverges as a power law: $P \approx r^\beta$. The best fit to our

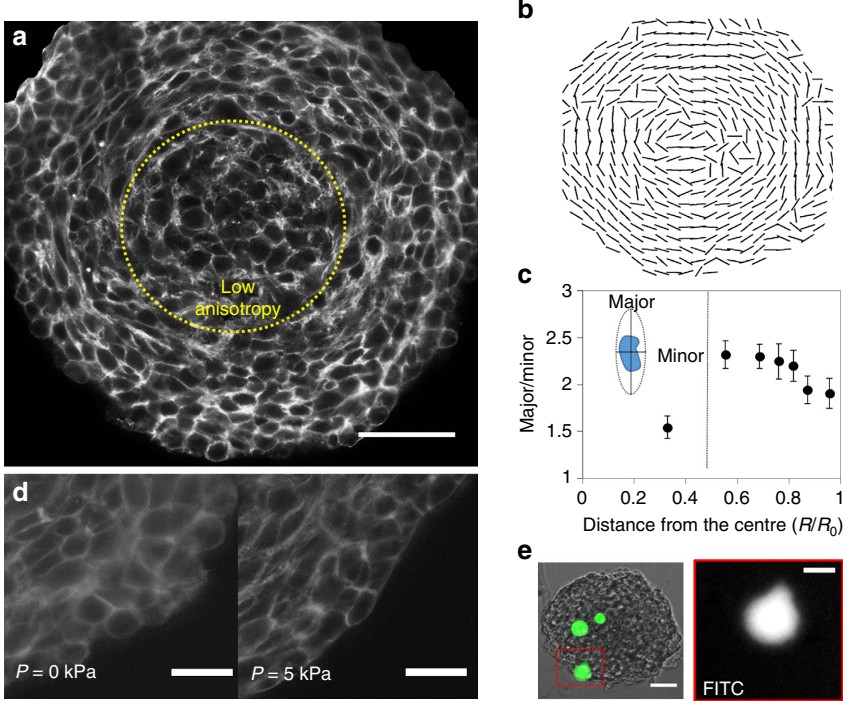

**Figure 5 | Organization of cells within CT26 spheroids.** (**a**) Confocal image of Phalloidin staining on CT26 spheroids at the equatorial plane. Yellow dotted circle divides a spheroid into a high anisotropy (major axis is perpendicular to radius; aspect ratio of 2–2.5) and low anisotropy zone (aspect ratio ~1.5). Scale bar, 50 µm. (**b**) Orientation of the major axis of cells at the equatorial plane. (**c**) Cell anisotropy has been estimated with the ratio of the major to the minor axis of the cell, as illustrated on the insert. Analysis of two axes has been performed using ImageJ and the fit ellipse function. Only cells with clearly marked cortical actin were used for the analysis. Data points ($N = 177$) were grouped together in bins, with the error bar being the s.e.m. and the position being the average $R/R_0$ position within the bin. The vertical line divides two zones: high anisotropy and low anisotropy zone. (**d**) Zoom-in on actin staining by the border of spheroids shows that cells at the surface are round and loosely bound, whereas after pressure is applied, the surface is well defined, and cells become more rectangular. Scale bar, 20 µm. (**e**) Bright-field/FITC image represents a spheroid with incorporated microbeads that are marked in green (FITC). Microbeads positioned at the surface of spheroids are highly deformed and represent a pear-like shape. Scale bars, 50 µm (bright field/FITC), and 20 µm (FITC).

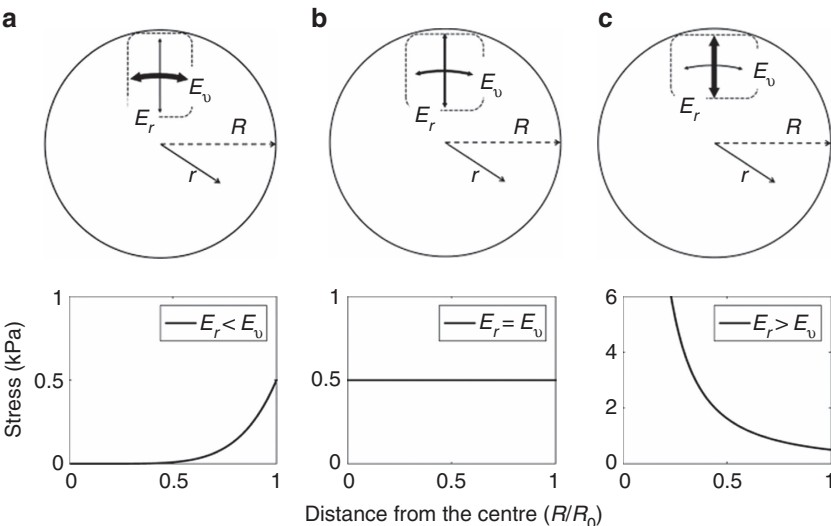

**Figure 6 | Stress distribution in dependence of the cellular anisotropy—theoretical model.** Graphical representation of the anisotropy within spheroids with (**a**) cells are softer in the radial direction ($E_r$) than tangentially ($E_v$), (**b**) cells are mechanically isotropic and (**c**) cells are stiffer in the radial direction. Below: radial stress profile along the radius computed with the theoretical model of Delarue et al. For (**a**) stress decreases towards the core, for (**b**) stress is constant, and for (**c**) stress increases towards the core.

data is obtained for $\beta = -0.21 \pm 0.1$ (mean ± s.e.), which corresponds to a 12% difference in between the radial and the ortho-radial stiffness of cells (ratio of $1.12 \pm 0.03$ (mean ± s.e.)). It has to be noticed that, whereas the anisotropic model captures the general radial stress profile, it does not explain the pressure-drop measured both at the spheroid centre and at its surface.

Non-disturbed proliferation of cells upon compression at the surface suggests an intrinsic response of cellular aggregates upon externally applied stress. Considering that the stress increase towards the core of the spheroid is due to cell anisotropy, one could expect in a spherical geometry, the anisotropy to vanish near the geometrical centre of the multicellular aggregate. Analysis of the cell shape at the equatorial plane confirmed transition from the region of high anisotropy to the region of low anisotropy within the core of the spheroid (Fig. 5c). This transition occurred at a distance of $r = R/3$. This value was much larger as compared with the cell size, but correlates well with the stress profile, which showed an equivalent transition approximately at the same distance from the centre.

## Discussion

Every important biological process in multicellular organism requires spatio-temporal control over key cellular decisions, such as proliferation or growth arrest. While for long it has been suspected that biochemical signalling plays a major role in defining the local cellular proliferation potential, it is now established that mechanical cues are equally important in morphogenesis and tissue homeostasis. Loss of homeostasis within tissue due to hampered environmental cues is a hallmark of many diseases, including cancer. Here, we presented a novel non-destructive biocompatible method to quantify the stress inside multicellular spheroids to understand how a compressive stress (physiologically occurring during tumour growth) is distributed within the 3D cellular aggregates, and if such stress is correlated with the cellular proliferation pattern.

Tumours for long were expected to have increased interstitial fluid pressure and tissue stress[14]. While knowledge about the interstitial fluid pressure in tumours has been widely studied, little is known about tissue stress distribution. One of the consequences of the elevated interstitial fluid pressure within tumours is significantly impeded transcapillary transport due to high vascular permeability and leakage[14,38]. Whereas decreased concentration of nutrients and oxygen within tumours could be potentially advantageous to limit tumour growth, unfortunately, drug intake and immune efficiency are hampered at the same time. One of the strategies in tumour treatment was therefore, use of antiangiogenic agents to improve the function of tumour vessels[39–41]. Despite the partial success, antiangiogenic agents cannot recapitulate proper function of tumour vessels which collapsed under accumulated tissue stress.

The vast majority of our knowledge on the tissue stress distribution within tumours comes from indirect observations. Stylianopoulos et al.[38,42] used freshly excised tumours, which were cut along the longest axis in $\sim 80\%$ of thickness, and deformation of the structure has been observed as a results of stress relaxation. The extent of the deformation has been measured and mathematical models were developed to estimate the tissue stress of excised tumours. Those experiments indicate that the accumulated stress during growth under constraint is not uniformly distributed, with the stress being zero at the surface[38]. By using full field optical coherence tomography (FFOCT) we have previously observed that the local strain (local displacement) under an isotropic stress was higher in the spheroid core than at the periphery[43]. Also, decreased cell-to-cell distance has been observed within the core upon isotropic compression[37]. Those indirect observations did not provide quantitative information on the stress distribution due to a lack of information on the local mechanical properties of the cells and their environment. Nevertheless, previous observations correlate well with our direct stress measurements that reveal

an increase of stress towards the core of the multicellular spheroids. Moreover, obtained pressure profile for CT26 spheroids can explain our previously observed differences in the biological response of cells at the surface from cells within the structure, in terms of the cell proliferation[33] and the cell cycle progression[19]. Incorporation of mechanically defined elastic cell-like sensors permitted for the first time local quantification of stress within 3D cellular aggregates. At the same time, we were able to relate the measured stress increase with the cellular shape anisotropy, which is potentially sufficient to promote non-homogenous stress distribution according to the theoretical model presented here. Our observations correlate well with the proliferation arrest of cells within the core[19,33].

Whereas the global volume reduction of the whole spheroid is compatible with a viscoelastic relaxation in response to the external stress, the increase of local stress field requires a more complex mechanical description of the spheroid. A theoretical framework has been proposed on the basis that active cellular rearrangements due to division or apoptosis dissipate mechanical stress, maintaining tissue in a liquid-like state[44]. In our experiments, we have observed that cells at the surface were continuously proliferating and actively rearranging under compressive stress, which could explain the pressure-drop at the surface. Following this hypothesis, increase of the stress towards the centre correlates with the observed hampered proliferation and as a consequence with a decrease of the rearrangement liberty (increase in jamming as confirmed with cell-to-cell distance measurements[37]) within the volume of the aggregates. Potential ability of tissue to dissipate stress is especially important in tumour research and requires further investigation, since lack of mechanical constraints gives a high invasive potential to cells at the surface of growing under constant compression tumours.

In conclusion, our presented methodology significantly improves direct 3D measurement of local stress. Fabricated PAA microbeads served as built-in pressure sensors to locally measure changes of the mechanical stress. Our approach is the first non-destructive method, which we validate by exploring the mechanical outcome of an applied compressive stress on multicellular spheroids. We anticipate a wide application of our microbead pressure sensors due their well defined and versatile mechanical properties (various rigidities) and choice of the functionalization ligand (that is, cadherin, integrins, ECM protein and so on). As we presented in this paper, the facility to incorporate microbeads within spheroids enables future application of automatized liquid robotic stations for high-throughput spheroid preparation. Therefore, our method can serve to perform high-throughput in vitro screenings of therapeutic agents on spheroids with additional information on the role of the stress. Moreover, owing to the biocompatibility of the PAA, we foresee application of our sensors in in vivo models, using microinjection or within xenografts, with particular interest in developmental biology, to evaluate the local isotropic pressure in embryos.

From an oncological perspective, we prove that the pressure increase generally observed in tumours can be reproduced in spheroids of cancer cells reconstituted in vitro. As compared with in vivo tumours, spheroids are extremely simple as they are (i) non-vascularized, (ii) composed of genetically and phenotypically identical cells and (iii) small enough to ensure the absence of necrotic core and limited heterogeneity of nutrients and oxygen supply. We postulate that such anomalous mechanical stress can be a general phenomenon, mainly due to the constitutive anisotropy of cells growing in an aggregate with spherical symmetry. In particular, this reappraises the role of fibroblasts and stromal compression as they appear not to be

essential to establish an internal overpressure in tumours. Moreover, a pressure-drop at the surface can explain lack of cell-contact inhibition and consequent continuous proliferation of cells at the surface, while the cell cycle of cells experiencing higher stress within the structure is stalled[19]. Finally our results suggest that the heterogeneity within tumours does not arise solely from the nutrients depletion and/or gradient of oxygen supply, but can be modulated by the growth-induced stress distribution.

## Methods

**Cell culture and spheroid formation.** Colon carcinoma cells CT26 (ATCC CRL-2638) were maintained in DMEM complete medium enriched with 10% fetal bovine serum (Life Technologies 61965-026) under 95% air and 5% $CO_2$ atmosphere.

Spheroids were prepared according to the classical agarose cushion protocol. Briefly, 100 μl of agarose (ultrapure agarose, Invitrogen, Carlsbad, CA) solution was dispensed per each well of 96-well plate and let to polymerize at 4 °C for 10 min. Subsequently, cells in suspension were seeded on top of the gel at a concentration of 300 cells per well for CT26. To accelerate formation of spheroids, plate with cell suspension were centrifuged for 5 min at 900 r.p.m. After 48 h spheroids present a shape close to a sphere.

**Application of compressive stress on spheroids.** We used osmotic stress to compress PAA beads and spheroids. It is exerted by supplementing the buffers and culture media with a well-defined amount of dextran, a large biocompatible polymer that is not internalized by eukaryotic cells. Dextran molecules (Sigma Aldrich, St Louis, MO) must be large enough to penetrate neither the polyacrylamide (PA) gel nor the MCS. We observed that dextran molecules with a molecular weight of 40 and 70 kDa do not enter the cells and polyacrylamide gels (Supplementary Note 1). For all experiments, we used 2 MDa dextran polymers (2 MDa, Sigma Aldrich, 95771) with a hydrodynamic radius of roughly 27 nm, to ensure high control over applied osmotic stress. Spheroids were imaged before and 30 min after the dextran has been introduced to the culture media. For Y27632-treated spheroids, spheroids were first imaged, then exposed to the drug at 10 μM concentration for 1 h, and subsequently dextran was added to media for 30 min followed by imagining.

**Spheroid fixation and actin staining.** Spheroids are fixed with 5% formalin (Sigma Aldrich, HT501128) in PBS for 30 min and washed once with PBS. Subsequently spheroids are exposed to tritonX-100 0.5% v/v (Sigma Aldrich, 93443) for permeabilization. Actin filaments were stained with phalloidin (Sigma, 147230). Glass cover slides were mounted on the glass slides with a Progold mounting medium (Life Technologies P36965) and stored at 4 °C before imaging.

**Fabrication of beads.** Polyacrylamide beads were fabricated using a water-oil emulsion approach. First a mix of acrylamide (Sigma Aldrich, 01697) and bisacrylamide (N,N′-methylenebisacrylamide, Sigma Aldrich, M1533) was prepared at the appropriate ratio (for ∼1 kPa gel 3/0.225, for ∼8 kPa gel 5/0.225, and for ∼40 kPa gel 8/0.48 with young's modulus defined by AFM). To initiate polymerization of 1 ml of acrylamide mix, we added 10 μl of APS (ammonium persulfate, at 10% w/v dissolved in PBS, Sigma Aldrich A3878), 0.75 μl of TEMED (Sigma Aldrich T9281) and 10 μl of acrylic acid (to improve EDC functionalization, Sigma Aldrich 147230). To form an emulsion of acrylamide mix in oil, we used HFE-7500 perfluorinated oil (3M, 98-0221-2928-5) with PFPE-PEG surfactant (kindly provided by Professor Garstecki). All solutions were degassed for 15 min in the vacuum chamber. The mix of polymerizing acrylamide and oil with surfactant has been vigorously shaken using a standard lab vortex (vortex genie-2) at maximal speed for 10 s. This way we formed droplets with diameter ranging from few micrometres up to 100 μm. To obtain emulsion of a submicrometer diameter, a higher-energy mixers such as Ultraturrex or high-pressure homogenizers would be required. Droplet solution has been purged with argon for 5 min and incubated under argon atmosphere in 60 °C for 1 h. To separate emulsion and transfer beads into PBS, we used perfluorooctanol (PFO, Sigma 370533). Briefly on the top of the emulsion, 500 μl of PBS were gently added. Subsequently, 400 μl of PFO was poured into the mix. After 2 min, by gentle pipetting the PBS phase, droplets were transferred into the water phase. Following the polymerization and transfer beads were washed thoroughly with PBS to remove excess of non-crosslinked acrylamide monomers. Beads were filtered on 40 μm cell drainer for incorporation within spheroids.

Beads were functionalized with fibronectin by EDC protein coupling. First beads were centrifuged in an ultra-low adhesive eppendorfs for 7 min at 2,000 r.p.m. and washed three times with MES buffer pH (ThermoFisher, 28390). Subsequently, beads were incubated with EDC (26 mg ml$^{-1}$, Sigma Aldrich E6383) in a MES buffer for 2 h under gentle agitation. After, beads were washed three times with MES buffer and incubated over night with fibronectin (Sigma Aldrich F1141) at the concentration of 40 μg ml$^{-1}$. To remove excess of fibronectin beads were washed six times with PBS.

**Characterization of beads.** We determined indirectly homogeneity of fabricated polyacrylamide beads by measurements of the diffusion time of small fluorescent molecules within the gel. For FCS we used a custom-built confocal microscope, which we already described in a previous article[45]. We used sulphorhodamine B (SRB, Sigma Aldrich 230162) as a sample molecule whose diffusion time in water at 23 °C is 27 μs. We determined the characteristic diffusion time of SRB in at least 30 PAA beads per each fabrication batch. The measurement was performed in a centre of the bead. Final diffusion time in a bead is a result of the fit of the autocorrelation function based on average of four runs of 20 s each.

To determine bulk modulus, beads were exposed to a number of osmotic pressures controlled by the concentration of dextran in the solution. At least 20 beads were used to determine compression detected by diameter decrease. We determined bulk modulus based on the region where deformation changes linearly with applied stress (below 10% strain).

**Immobilization of beads within the spheroids.** Functionalized fluorescent beads were added to the single-cell suspension before seeding on agarose gels. The concentration of beads has been estimated by calculation of the number of beads in a 1 μl volume drop. We have added around 5–10 beads per well since during centrifugation process some beads remain outside the region occupied by cells and not all of the beads are incorporated within the forming structures.

**Image acquisition and processing.** We used a Leica inverted confocal microscope (DM-IRB; Leica Microsystems, Bannockburn, IL) equipped with PMT detectors. To observe 3D organization of cells within spheroids and around incorporated beads we used a X40 oil objective (numerical aperture 1.3).

To measure compression of beads within spheroid, we used an inverted Nikon Eclipse Ti microscope, equipped with mercury fluorescent lamp (Nikon Intensilight), motorized table with controlled temperature and $CO_2$ level (LCI CH-109; FC-5). We used ×10 objective (numerical aperture 0.3) with 1.5 optical multiplier. In our experiments we used Andor Lucas and widefield Andor Neo cameras. Microscope has been operated using μmanager freeware. Images of epifluorescence were taken at the equatorial plane of the bead, which was determined manually at the point where focus of the bead was the best. Similarly equatorial plane was determined for spheroids.

**Image analysis.** To define surface of beads at the equatorial plane we have used automated isoData threshold. For more details please refer to Supplementary Information.

**Data availability.** The data that support the findings of this study are available from the corresponding author upon reasonable request.

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

## Acknowledgements

We thank D. Centanni for discussions on microfabrication technologies and J.-F. Joanny for his detailed explanation of the theoretical model we have presented in this manuscript. This work was supported by the Agence Nationale pour la Recherche (Grant ANR-13-BSV5-0008-01), by the Institut National de la Santé et de la Recherche Médicale (Grant 2011-1-PL BIO-11-IC-1).

## Author contributions

M.E.D. and G.C. designed research; M.E.D. performed experiments; M.D. contributed to results in Supplementary Information; F.I. and A.D. contributed to experiments on polyacrylamide calibration; M.E.D. and G.C. analysed data; J.P. theoretical model contribution; M.E.D. and G.C. wrote the manuscript.

## Additional information

**Competing financial interests:** The authors declare no competing financial interests.

