## [Peer Review File · Nature Communications]

Reviewers' comments:

Reviewer #1 (Remarks to the Author):

This is a highly interesting reports of a novel method to measure local micron-scale pressures within 3-D cellular aggregates. Application of this method to cell spheroids reveals interesting non-uniform pressure distributions that correlate with structural differences in cells and have implications for control of cell growth in tissue and tumors. Several issues need to be addressed to increase clarity and impact of the report.

I. 38. Which mechanical properties re referred to here?

I. 40 not clear what "normally malignant cells" means

I. 116 quantify

I.141 not clear what this means. " to take into account the non-linearity appearing for a compression yield exceeding 15%." Should be defined or explained.

I. 145. Non-isotropic deformations of the bead also induce shear. Does the shear modulus of the gel formulation (in bulk, not necessarily in beads) correspond to what is measured for bulk modulus?

I. 154 Why would a necrotic core could drastically hamper measurements? It would seem to make the setting more like some aspects of a real tumor

I. 167 re "Deformed beads were eliminated from analysis" how is "deformed" defined. It would seem that deformation is what is needed to compute forces. Why shouldn't spheroids be able to apply force to the bead before the osmotic pressure was applied? What fraction of beads in spheroids counted as deformed?

I.202 More quantification or explanation is needed for the claim that that the stress in the whole spheroid never reached that of the external pressure. The strain from external pressure was 23% and in the core was 25%?

I.226 this is fig 5E not SI

I. 253 it could also be that the stress at the surface is not enough to perturb proliferation

I. 267 not clear what specifically is meant by "Loss of such functional cellular pattern within tissue ..." which pattern?

I.273 reference is needed

I. 291 how was local strain (local displacement) measured?

Reviewer #2 (Remarks to the Author):

Some questions are as follows:

1. An important content in this manuscript is preparation of the non-destructive cell-like microsensors (PAA beads). The properties of PAA beads will affect the observation results. Thus, the authors should describe in more detail the preparation process and properties of bead. For example, How did the temperature affect the PAA beads properties, including beads size distribution and pore structure in bead?
2. In the fabrication of PAA microbeads, the authors used an oil-in-water emulsion approach. How was vortexing formed? What rotation speed is controlled? In normal, this method will prepare beads with large distribution of size and the rough surface appearance of beads. Further more, in this work, not only size distribution but also surface appearance of beads showed good results. Please add a photo on the surface appearance and pore size within bead.
3. The authors found that the fabrication method was stable and only little sensitive to the concentration of surfactant. The question is how the method is sensitive to the kinds of surfactant?

Reviewer #3 (Remarks to the Author):

Dolega et al in their manuscript present a clever method to determine intratumoral pressure in physiologically relevant 3D spheroid assays. They achieved this by fabricating microsensors comparable in size to cells within thick tissue of defined elasticity. Using this method they determined that anisotropic order of cells within the microtissue is sufficient to explain transference of pressure within the tissue. They determined that this was exclusive of proliferation within the tissue. This method is needed as the field is still struggling to decouple physical and chemical effects on cell fate as they relate to malignancy and normal tissue homeostasis. Moreover, in the context of normal tissue maintenance vs. cancer promotion on length scales and timescales that is relevant for single cells in thick organs within a living organism. A major strength of this paper is the link with the mechano-osmotic coupling to determine the stress-strain behavior of these probes.

As it stands this method gives a quantitative method to determine pressure within tissue. Some more statistical analysis will drive the comparison home more. It would benefit from some additional experiments to make this more palatable for a non-physics oriented audience. I think the following experiments/ suggestions should be incorporated to make this manuscript stronger.

Major concerns:

- 1) Abstract states that this assay uses biologically homogeneous samples but the microtissues were derived by aggregating single cells that may show both genetic and phenotypic variability. If this is the point the authors are trying to make, please derive aggregates from a single cell using clonal expansion to form the microtissue.
- 2) These measurements were made with tumorigenic cells, please perform with microtissues derived from normal tissue, preferably with a normal matched tissue to the ones used if available.
- 3) Please use a live cell dye to localize membrane to see resultant cell shape changes with or without applied pressure.
- 4) If there is no observable proliferation, can you determine what stage of the cell cycle?
- 5) If cell shape is the main focus, can you change this parameter by altering either targets if rock or rac that have been shown to regulate cell shape in the presence and absence of pressure.
- 6) Can we link some biological change to the applied pressure? This assay can be powerful if this is elucidated.

Minor points

A few spelling errors that spell check will catch

Reviewer #1 (Remarks to the Author):

This is a highly interesting reports of a novel method to measure local micron-scale pressures within 3-D cellular aggregates. Application of this method to cell spheroids reveals interesting non-uniform pressure distributions that correlate with structural differences in cells and have implications for control of cell growth in tissue and tumors. Several issues need to be addressed to increase clarity and impact of the report.

We thank reviewer #1 for these positive comments.

I. 38. Which mechanical properties are referred to here? & I. 40 not clear what "normally malignant cells" means

We have changed the text to eliminate this ambiguity

In line 38:

"Extensive in vitro studies on the mechanical cues (i.e. ECM rigidity, application of a flow to induce shear stress), showed that these alone can promote malignant phenotype in a non-malignant cells¹ or promote proper 3D growth and development of malignant cells²."

I. 116 quantify

We have corrected the text.

I.141 not clear what this means. " to take into account the non-linearity appearing for a compression yield exceeding 15%." Should be defined or explained.

We have added the following sentence to clarify the stress/strain relation and define the linear and non-linear elastic regions. Line 141

"In an elastic region of deformation (compression below 15%) the stress is linearly proportional to the strain. To model mechanical properties of polyacrylamide beads out of the linear regime, we used an empiric polynomial Mooney-Rivlin model. "

I. 145. Non-isotropic deformations of the bead also induce shear. Does the shear modulus of the gel formulation (in bulk, not necessarily in beads) correspond to what is measured for bulk modulus?

For the used formulation the expected Young's modulus is 8.44 ± 0.82^3 kPa. This corresponds to the bulk modulus we measured for microbeads (14 kPa) for a Poisson's ration of 0.4.

I. 154 Why would a necrotic core could drastically hamper measurements? It would seem to make the setting more like some aspects of a real tumor

The presence of a necrotic core drastically hampers imaging, as spheroids become less transparent and auto-fluorescence of cell debris in the necrotic core appears. We have avoided using spheroids with a necrotic core also because we are unable to characterize this region of dead cells in term of its composition (it contains a mix of wreckage and interstitial fluid, the viscosity of which is unknown to us). This implies a lack of a direct contact with cells and makes the results difficult to interpret within our theoretical framework.

I. 167 re "Deformed beads were eliminated from analysis" how is "deformed" defined. It would seem that deformation is what is needed to compute forces. Why shouldn't spheroids be able to apply force to the bead before the osmotic pressure was applied? What fraction of beads in spheroids counted as deformed?

Deformed beads were the beads of the visibly elliptical shape. Because the fraction of deformed beads before any stress has been applied was very small (approximately 1/100) we assumed that these beads were deformed during preparation process (i.e. stacked on the wall of the tube) (please check Figure 1). We agree with the referee that spheroids can impose a pre-constraint on beads that we cannot quantify. Thus we concluded in this manuscript on the pressure propagation upon externally applied stress. Ability to quantify the absolute value of local stress within tissue is a next step in the improvement of the technique.

In order to provide more information concerning this point we have changed the text as follows (Line 169):

“In rare cases we have encountered beads that were initially deformed into an elliptical shape (a fraction of approximately 1/100). We eliminated these beads from the analysis due to their occasional appearance and lack of clear explanation of their origin (i.e. during the fabrication process).”

Figure 1 Epi-fluorescence image of fabricated microbeads in DPBS. Left panel presents the population of FITC-beads of different size with few (marked by yellow circles) representing the “elliptical” shape. Right panel is a zoom in to show more clearly the shape of deformed beads. Scale bar 50 μm .

I.202 More quantification or explanation is needed for the claim that that the stress in the whole spheroid never reached that of the external pressure. The strain from external pressure was 23% and in the core was 25%?

We thank referee for pointing out this ambiguity. The local average strain measured inside spheroid has increased up to 22.4% and not 25% as stated by mistake in the manuscript. Of course, single beads may be more compressed than expected for an additional pressure of 5 kPa. To correct the statement and provide a better explanation we have introduced following changes in line 204:

“On the contrary, the strain progressively increased towards the spheroid core indicating that in the whole volume of the spheroid, measured stress was significantly lower than the externally applied pressure. Only by 30% of the distance from the core (r/R_0) measured mean strain was very close (and locally exceeded) to the one obtained for isolated beads under 5kPa pressure.”

I.226 this is fig 5E not SI

We have corrected the figure reference.

I. 253 it could also be that the stress at the surface is not enough to perturb proliferation

Yes. We have worked on the role of the compressive stress on multicellular spheroids, and we have seen a non-homogenous cellular response to applied pressure. Here we have shown for the first time that cells at the surface are subjected to a smaller pressure than cells within the structure, and this further correlates with previously observed differences in biological response in the cell proliferation⁴ and the cell cycle progression⁵.

To stress on this point we have added following sentence (line 310):

“Moreover, obtained pressure profile for ct26 spheroid can explain our previously observed differences in the biological response of cells at the surface from cells within the structure, in terms of the cell proliferation³³ and the cell cycle progression¹⁹”

I. 267 not clear what specifically is meant by "Loss of such functional cellular pattern within tissue ..." which pattern?

To simplify the message we have changed the text as follows (line 279):

“Loss of homeostasis within tissue due to hampered environmental cues is a hallmark of many diseases, including cancer”

I.273 reference is needed

We have added the following reference within the text:

Y. Boucher et al, Cancer Research, 1990 ; (50)15; 4478-4484

I. 291 how was local strain (local displacement) measured?

We have added a reference to the publication in which we have used an FFOCT (full field optical coherence tomography) which allowed us to directly obtain an image at the equatorial plane of the immobilized spheroids. Since the technique is rapid and can be used directly on non-fixed samples, we have obtained images before and after pressure has been applied. The local strain has been defined from the local displacement of the characteristic to the image zones.

We have changed the text to clarify the statement and the corresponding references. (Line 303)

“By using full field optical coherence tomography (FFOCT) we have previously observed that the local strain (local displacement) under an isotropic stress was higher in the spheroid core than at the periphery⁴³. Also, decreased cell to cell distance has been observed within the core upon isotropic compression⁶.”

Reviewer #2 (Remarks to the Author):

1. An important content in this manuscript is preparation of the non-destructive cell-like microsensors (PAA beads). The properties of PAA beads will affect the observation results. Thus, the authors should describe in more detail the preparation process and properties of bead. For example, How did the temperature affect the PAA beads properties, including beads size distribution and pore structure in bead?

We agree with the referee that properties of PAA beads will affect the observations results. We have elaborated our protocol in order to produce homogenous beads (in terms of mechanical properties) in a robust and reproducible manner. For our studies the crucial parameter was the bulk modulus which was obtained after polyacrylamide beads were polymerized and thoroughly washed with DPBS. In this way we were ensured of mechanical properties (bulk modulus) of beads used for the measurements performed within spheroids. Such strategy allowed translating observed strain into a measure of the local pressure.

We chose to use PAA as a material for pressure sensors because it has been broadly studied in terms of fabrication/polymerization (role of the temperature, presence of oxygen, concentration of bisacrylamide/acrylamide, etc...) and extensively characterized for *in vitro* applications (i.e. traction force microscopy etc..). Therefore, in the manuscript we have determined only which parameters play crucial role in the polymerization process and which of them have direct influence on the "structural" homogeneity of the batch.

We agree with the Referee that the temperature can have an effect on the polymerization of PAA. On the one hand, it has been shown that the temperature affects the structure of the gel, making it less transparent and more rigid ⁷, (Pruitt et al 2016). On the other hand, temperature affects the emulsion formation through alteration of surface tension. Even if we could change the temperature we were never able to separate these two effects and therefore, we decided to fix a temperature for each step of batch preparation. In all our experiments we prepared gels at RT and polymerized gels at 60°C which rendered compressible microbeads with a bulk modulus of a proper range and a size distribution allowing to isolate beads of size being similar to a single cell.

2. In the fabrication of PAA microbeads, the authors used an oil-in-water emulsion approach. How was vortexing formed? What rotation speed is controlled? In normal, this method will prepare beads with large distribution of size and the rough surface appearance of beads. Furthermore, in this work, not only size distribution but also surface appearance of beads showed good results. Please add a photo on the surface appearance and pore size within bead.

The following changes have been applied to the materials and methods (Line 403):

"The mix of polymerizing acrylamide and oil with surfactant has been vigorously shaken using a standard lab vortex (vortex genie-2) at maximal speed (3200 rpm) for 10 seconds. This way we formed droplets with diameter ranging from few micrometres up to 100 µm. To obtain emulsion of a submicrometer diameter, a higher-energy mixers such as Ultraturrex or high-pressure homogenizers would be required."

Measuring the pore size is not straightforward and in most instances remains qualitative. Observation of the gels structure by scanning electron microscope requires to dehydrate the

gels which has a direct influence on the subsequently observed pore size (Please check the image of SEM included below as figure 2).

Figure 2 SEM image of a section of a PAA microbead.

Therefore, instead of measuring pore size within beads we have determined the exclusion size by observation of which large fluorescent polymers of known hydrodynamic radius infuse into the volume of the beads. We have estimated the exclusion pore size to be in the range of 1 – 9 nm (Supplementary Information 1).

We have added to figure 1 a fluorescence image of beads and a fluorescence image of the fibronectin surface coating of PAA microbeads. The new Figure 1 is as follows:

Figure 1 Characterization of Polyacrylamide (PAA) beads. A) Brightfield image of polymerized PAA beads after filtration. Scale bar: 50 μm . B) Fluorescence image of PAA microbeads containing trapped large polymers functionalized with FITC; scale bar 50 μm . C) Fluorescence image of coating of PAA beads with Cy3-Fibronectin; scale bar 50 μm . D) Distribution of size of polyacrylamide beads in dependence on the concentration of PFPE-PEG surfactant (1%, 3%, and 5%) during initial vortexing. E) Characteristic diffusion time of SRB molecules, using FCS, within gels indicated uniformity of beads (small dispersion) and mechanical properties. Small time of diffusion is characteristic for soft gels and increases with the volume fraction. Gels are defined by the acrylamide/bisacrylamide ratio. On the right, for 5/0.225 gels we show the effect of mixing on the uniformity of the batch. Each point corresponds to a single measure of the diffusion time within the bead; F) Uniformity and reproducibility is maintained

between batches (sample 1, sample 2, and sample 3) of the same acrylamide/bisacrylamide ratio (5/0.225). E)-F) mean +/- SD

3. The authors found that the fabrication method was stable and only little sensitive to the concentration of surfactant. The question is how the method is sensitive to the kinds of surfactant?

Surfactants play a major role in the emulsification process by stabilizing the dispersed phase in the continuous phase. To produce a water-in-oil emulsion a careful choice of the oil and surfactant is required in order to provide a system that is biocompatible and stable. For a broad choice of oils (silicone oils, perfluorinated oils or hydrocarbon oils) there is a list of surfactants that will be compatible to stabilize water droplets⁸. Initially we chose one of the most common systems used in microfluidics composed of mineral oil and SPAN 80 as a surfactant. However, since the mineral oil has a smaller density than water, the transfer of beads after polymerization was inconvenient. Beads had to be washed (pipetted up and down) with mineral oil that does not contain the surfactant in order to force the transfer. After each single step of washing emulsion had to be centrifuged to remove the mineral oil and replace it with fresh. The viscosity and the fact that the oil phase was always beneath the water phase was experimentally very impractical and did not provide very good results (Figure 3). Therefore, we have decided to use a perfluorinated oil due to its low viscosity (0.77 cSt as provided by the manufacturer), and high density (1614 kg/m³ as provided by the manufacturer). Moreover, transfer of beads is induced chemically by addition of another perfluorinated molecule making the protocol much simplified. One of the drawbacks of course is the capability to dissolve large quantities of oxygen, and that's why (as explained in the manuscript) we have degassed all the solutions and purged them with argon.

Figure 3 Effect of the oil/surfactant on the polymerization/fabrication of microbeads.

Reviewer #3 (Remarks to the Author):

Dolega et al in their manuscript present a clever method to determine intratumoral pressure in physiologically relevant 3D spheroid assays. They achieved this by fabricating microsensors comparable in size to cells within thick tissue of defined elasticity. Using this method they determined that anisotropic order of cells within the microtissue is sufficient to explain transference of pressure within the tissue. They determined that this was exclusive of proliferation within the tissue. This method is needed as the field is still struggling to decouple physical and chemical effects on cell fate as they relate to malignancy and normal tissue homeostasis. Moreover, in the context of normal tissue maintenance vs. cancer promotion on length scales and timescales that is relevant for single cells in thick organs within a living organism. A major strength of this paper is the link with the mechano-osmotic coupling to determine the stress-strain behavior of these probes. As it stands this method gives a quantitative method to determine pressure within tissue. Some more statistical analysis will drive the comparison home more. It would benefit from some additional experiments to make this more palatable for a non-physics oriented audience. I think the following experiments/ suggestions should be incorporated to make this manuscript stronger.

We thank reviewer for these very positive comments and suggestions.

Major concerns:

1) Abstract states that this assay uses biologically homogeneous samples but the microtissues were derived by aggregating single cells that may show both genetic and phenotypic variability. If this is the point the authors are trying to make, please derive aggregates from a single cell using clonal expansion to form the microtissue.

We agree with the referee that the sentence within the abstract about the use of biologically homogenous samples is misleading. To clarify this point we have modified the statement in the abstract:

“This observed pressure profile is explained by the anisotropic arrangement of cells and our results suggest that such anisotropy alone is sufficient to explain the pressure rise inside spherical aggregates composed of a single cell type.”

2) These measurements were made with tumorigenic cells, please perform with microtissues derived from normal tissue, preferably with a normal matched tissue to the ones used if available.

Comparison of the pressure profile of the normal and malignant spheroids is an important point but states as a whole new research project. In our manuscript we decided to base our research on the ct26 cell line because we have been working on it since the original questions of the role of the compressive stress appeared. Unfortunately there is no corresponding “normal” cell line available for CT26. We agree on the strong interest of this comparative approach and consider the suggestion for a future work, by choosing a pair of corresponding cell lines (malignant and non-malignant).

3) Please use a live cell dye to localize membrane to see resultant cell shape changes with or without applied pressure.

Microscopic observation of cellular aggregates induces multiple optical aberrations, and thus live-imaging is an obstacle. We have initially tried to perform observation of the cellular membrane live by using stable Actin-GFP CT26 cell line. While the signal of single cells *in vitro* was very satisfactory we have never obtained good resolution images within spheroids

(only first two layers of cells at the equatorial plane were visible with a poor resolution). We have experimentally observed that for small spheroids (100-200 μm) we are able to obtain Z-section of only first 50 μm of the structure regardless the microscopic technique used (confocal or two-photon microscope). Indeed, in the literature, the vast majority of the analysis and immunolabeling is performed on fixed cryosections of spheroids in order to reach the core of the structure or a maximal projection is presented instead. The best results obtained so far are with light sheet microscopy, however, it is impossible to perform dynamic studies on the spheroid compression due to technical limitations (immobilization of spheroids within agarose inside a capillary).

Instead of using live cell dye we have introduced to a surrounding medium a fluorescent dye. Sulphorhodamine B (SRB) is a polar dye which does not penetrate the cellular membrane but diffuses in the extracellular space of spheroids. Since the concentration of the dye can be freely varied without any harm to the cells we have used this method to observe the cell shape on live samples before and after compressive stress has been applied. However, we were still limited by the imaging depth to 50 μm (Figure 4). We have discussed with other researchers working on spheroids (Jean-Claude Vial – optics for tissue biology, Helene Delanoe - biophysics of spheroids, Pierre Nassoy – biophysics of spheroids), and we were assured about the difficulties related to the live imaging.

Figure 4 Z-stack of the CT26 spheroids without any compressive stress. Each image corresponds to a Z-section in the plane of 6, 18, 30, 42, and 54 μm respectively. Cells are easily distinguishable until the depth of 30 μm , and the information about the extracellular space is lost around the plane at 50 μm .

Therefore, the images presented in Figure 4 obtained from live observation, were not taken at the equatorial plane and thus we were unable to perform the same quantification as for the fixed samples. We performed a time-lapse observations of the global form of the spheroids at 50 μm during the first 60 minutes after compressive stress has been applied. We have observed a rearrangement of cells at the surface of the structure as shown on figure 5. We also noticed that the extracellular space network (marked with the SRB) became sharper suggesting that the application of the compressive stress is associated not only with cells rearrangements but might be also linked to a global compression of a soft extracellular matrix that exists between cells.

Figure 5 Observation of the cell rearrangements after application of the compressive stress. Left panel) represents the same spheroid before the dextran has been added and after 60 minutes under compression. Right panel) represents the time-lapse images taken every 5 minutes.

4) If there is no observable proliferation, can you determine what stage of the cell cycle? & 6) Can we link some biological change to the applied pressure? This assay can be powerful if this is elucidated.

We find question 4 and 6 directly correlated and therefore, we would like to join them together to provide a comment.

We have shown previously that the application of compressive stress to spheroids is correlated with the change of a pattern of proliferating cells within the spheroid. Cells from the border continued to proliferate whereas cells in the core underwent the cell cycle arrest at the restriction point^{4,5}. Moreover, in here we have shown that this proliferation pattern along the radius corresponds to the pressure profile observed. Low pressure was measured for proliferating cells at the border, and significantly higher pressure has been observed within the volume of the spheroid, where as previously shown, cells stopped dividing when exposed to a compressive stress. Such change in observed proliferation is a good example of a “biological change”.

We have shown that application of compressive stress blocks the cell cycle at the late G1 check point, also called the Restriction point⁵. Western blot analysis showed a progressive decrease of the repressor protein pRb level with time and corresponding to an increase of the kinase inhibitor p27^{kip1}. We showed also that the expression of other cyclins (D1, E) and kinase inhibitor (p21^{Cip1}) is not affected by the compressive stress.

We have added this information in the text in the line 214:

“Our previous observations revealed that the compressive stress inhibits cell proliferation by an overexpression of the kinase inhibitor p27^{kip1} at the level of the Restriction point⁵.”

5) If cell shape is the main focus, can you change this parameter by altering either targets of rock or rac that have been shown to regulate cell shape in the presence and absence of pressure.

Following the reviewer suggestion we decided to target ROCK, which in non-muscle cells controls actin-cytoskeleton assembly and cell contractility. We used a Y-27632 inhibitor of ROCK in order to study the importance of the cytoskeletal activity on the propagation of the pressure within spheroids exposed to a compressive stress.

Initially we verified how addition of the Y27632 drug (at 10 μM) influences the growth of spheroids, and if the treatment changes the global compression of the spheroids. We observed that addition of Y27632 has had a direct and almost immediate effect on the growth of spheroids. In the timescale of our experiments ROCK inhibition has stopped spheroids growth and the volume of the spheroid was not increasing (during 1 hour of observation after the drug has been added) in contrast with the control spheroids, which continued to grow (Figure 6a). Interestingly, inhibition of ROCK did not change the global spheroids compression, which was 8.9% for a control, and 8.8% for Y-27632 treated spheroids (non-significant difference, $N=50$ control and $N=53$ for Y27632, $p=0.9639$) (Figure 6b).

Figure 6 **A**) Evolution of spheroids size during 1 hours after treatment with Y27632 and in a control ($N=8$ spheroids for Y27632, and $N=7$ for the control). Error bar \pm SD **B**) Comparison of a global compression of spheroids non-treated ($N=50$) and treated with 10 μM Y27632 ($N=53$). Error bar \pm SEM. Statistical test – t-test with $p=0.0639$. **C**) Measured strain in the function of the position within the spheroid (0 –core and 1 - border of the spheroid) for spheroids treated with Y27632 (blue, $N=135$) and for the control (red, $N=81$). Data points are grouped together in bins, with the error bar being a S.E.M and the position being an average position within the bin.

Within the given time, we performed multiple experiments on the propagation of pressure externally applied to spheroids treated with Y27632 drug. New experiments were associated with the fabrication of a new batch of a corresponding bulk modulus ($11 \text{ kPa} \pm 2$) and the adjustment of the protocol to include the drug treatment. We have also performed series of control experiments with the new batch of beads to enforce the statistics (3 separate experiments for each condition; control $N=81$; Y27632 $N=135$). Figure 6C presents the change of measured beads strain along the radius upon spheroids compression. For the new control experiments we have observed that the strain by the border (distance from the center 0.6-1) is lower than expected for the applied pressure, and increases towards the core, as has been observed and described previously in the manuscript. Interestingly, in ROCK-inhibited spheroids, the strain was noticeably higher at the border and achieved levels comparable to the control by the center of spheroids. Our results point out that ROCK inhibition had visible influence on cells located at the border, which as shown previously, were the ones that were biologically active and continued to proliferate in the growing spheroids. ROCK inhibition had

also a direct influence on the spheroids growth and thus, our results further suggests that proliferation itself can be responsible for the accumulation of cellular anisotropy within the spheroids (as previously proposed⁶), which in turn has an influence on the propagation of the pressure.

In presented results we have obtained strains that are higher than those observed previously. In the center of spheroids we have regularly measured strain of $\sim 0.3-0.4$ for all repeated control experiments as well as for treated with Y27632 spheroids. Of course, as compared to previous experiments multiple parameters have changed (batch of cells, batch of beads, protocol etc). However, our improved protocol of beads functionalization with fibronectin might be responsible for an increased cell-to-bead interaction. As cells in tissues are under tension, it is possible that beads at their initial state are extended, and thus strain is larger when spheroids are compressed. We believe that this observation opens new questions on the role of tension and cellular contractility on the response to stress, and those questions require future investigation.

We propose to insert our results as a supplementary material because the presented theoretical model does not take into account presence of tension between cells and also because this results still remain preliminary and demand further work. However, following the reviewers comments, we continue to orient our research towards the profound understanding of the role of the cytoskeleton and its dynamics on the pressure propagation.

We have changed the text of the manuscript in line 234 to introduce experiments performed with Y27632 drug:

“Several studies confirm the role of the organization of the cytoskeleton components and polarity on the anisotropic response to stress. Such structural anisotropy may lead to anisotropy in the mechanical response of the cells. We have inhibited ROCK, which is known to control cell cytoskeleton assembly and contractility, and we observed that pressure propagation was altered as compared to control (Supplementary Information 5). Interestingly, observed difference occurred in the region where cells in CT26 spheroids continue to divide. This suggests that the proliferation might be one of the factors imposing anisotropy within the cellular aggregates. Moreover, cells shape quantification revealed a discontinuity as the anisotropy decreased in the core of the aggregate (aspect ratio ~ 1.5), where cells become rounder (Figure 5A and 5C).”

We have also detailed the protocol of drug treatment in Materials and Methods section.

References

- 1 Butcher, D. T., Alliston, T. & Weaver, V. M. A tense situation: forcing tumour progression. *Nature Reviews Cancer* **9**, 108-122, doi:10.1038/nrc2544 (2009).
- 2 Weaver, V. M. *et al.* Reversion of the malignant phenotype of human breast cells in three-dimensional culture and in vivo by integrin blocking antibodies. *Journal of Cell Biology* **137**, 231-245, doi:10.1083/jcb.137.1.231 (1997).
- 3 Tse, J. R. & Engler, A. J. Preparation of hydrogel substrates with tunable mechanical properties. *Current protocols in cell biology / editorial board, Juan S. Bonifacino ... [et al.] Chapter 10*, Unit 10.16-Unit 10.16, doi:10.1002/0471143030.cb1016s47 (2010).
- 4 Montel, F. *et al.* Isotropic stress reduces cell proliferation in tumor spheroids. *New Journal of Physics* **14**, doi:10.1088/1367-2630/14/5/055008 (2012).
- 5 Delarue, M. *et al.* Compressive Stress Inhibits Proliferation in Tumor Spheroids through a Volume Limitation. *Biophysical Journal* **107**, 1821-1828, doi:10.1016/j.bpj.2014.08.031 (2014).
- 6 Delarue, M., Joanny, J.-F., Juelicher, F. & Prost, J. Stress distributions and cell flows in a growing cell aggregate. *Interface Focus* **4**, doi:10.1098/rsfs.2014.0033 (2014).
- 7 Chen, B. & Chrambach, A. ESTIMATION OF POLYMERIZATION EFFICIENCY IN THE FORMATION OF POLYACRYLAMIDE-GEL, USING CONTINUOUS OPTICAL-SCANNING DURING POLYMERIZATION. *Journal of Biochemical and Biophysical Methods* **1**, 105-116, doi:10.1016/0165-022x(79)90017-4 (1979).
- 8 Baret, J. C. Surfactants in droplet-based microfluidics. *Lab on a Chip* **12**, 422-433, doi:10.1039/c1lc20582j (2012).

REVIEWERS' COMMENTS:

Reviewer #1 (Remarks to the Author):

The authors have made reasonable responses to the initial review and the conclusions made originally are now better supported by data on ROCK inhibition. I think this work will be of interest to many readers.

Reviewer #2 (Remarks to the Author):

The authors have given clear responses to three reviewers on their manuscript. At same time for me, the authors have explained the beads formation, properties and effect on cells, I accept their explanation.

And also the authors revised this manuscript according to three comments.

Reviewer #3 (Remarks to the Author):

The authors have sufficiently addressed my concerns